# New Insights into the Clinical Implications of Yes-Associated Protein in Lung Cancer: Roles in Drug Resistance, Tumor Immunity, Autophagy, and Organoid Development

**DOI:** 10.3390/cancers13123069

**Published:** 2021-06-20

**Authors:** Geon Yoo, Dongil Park, Yoonjoo Kim, Chaeuk Chung

**Affiliations:** 1Clinical Research Division, National Institute of Food and Drug Safety Evaluation, Cheongju 28159, Chungcheongbuk-do, Korea; geonyoo@korea.kr; 2Division of Pulmonology and Critical Care Medicine, Department of Internal Medicine, College of Medicine, Chungnam National University, Daejeon 35015, Korea; rahms@cnuh.co.kr (D.P.); k1206j@cnuh.co.kr (Y.K.); 3Infection Control Convergence Research Center, Chungnam National University School of Medicine, Daejeon 35015, Korea

**Keywords:** YAP, TAZ, Hippo pathway, lung cancer, drug-resistance, EGFR-TKI, PD-L1, autophagy, organoid

## Abstract

**Simple Summary:**

Innovative advancements in lung cancer treatment have developed over the past decade with the advent of targeted and immune therapies. Yes-associated protein (YAP), an effector of the Hippo pathway, promotes the resistance of these targeted drugs and modulates tumor immunity in lung cancer. YAP is involved in autophagy in lung cancer and plays a prominent role in forming the tubular structure in lung organoids and alveolar differentiation. In this review, we discuss the central roles of YAP in lung cancer and present YAP as a novel target for treating resistance to targeted therapies and immunotherapies in lung cancer.

**Abstract:**

Despite significant innovations in lung cancer treatment, such as targeted therapy and immunotherapy, lung cancer is still the principal cause of cancer-associated death. Novel strategies to overcome drug resistance and inhibit metastasis in cancer are urgently needed. The Hippo pathway and its effector, Yes-associated protein (YAP), play crucial roles in lung development and alveolar differentiation. YAP is known to mediate mechanotransduction, an important process in lung homeostasis and fibrosis. In lung cancer, YAP promotes metastasis and confers resistance against chemotherapeutic drugs and targeted agents. Recent studies revealed that YAP directly controls the expression of programmed death-ligand 1 (PD-L1) and modulates the tumor microenvironment (TME). YAP not only has a profound relationship with autophagy in lung cancer but also controls alveolar differentiation, and is responsible for tubular structure formation in lung organoids. In this review, we discuss the various roles and clinical implications of YAP in lung cancer and propose that targeting YAP can be a promising strategy for treating lung cancer.

## 1. Introduction

The Hippo signaling pathway has been one of the most actively studied pathways in biomedicine over the past decade [1,2,3]. It was first discovered in Drosophila, where it regulates cell differentiation and proliferation during development [4,5]. Studies in mice later revealed that it is highly involved in tumorigenesis and metastasis [2]. Yes-associated protein (YAP) is a downstream transcriptional co-activator of the Hippo pathway. It controls cell proliferation and apoptosis by regulating the transcription of genes that control DNA replication, DNA repair, apoptosis, and metabolism [6]. YAP also cooperates with other transcription factors, including AP1, β-catenin, and cytokines such as TGF-β, to control many cellular functions [3,7]. Furthermore, it senses and responds to mechanical stress [8]. Mechanotransduction signals related to matrix stiffness directly influence the subcellular localization and activity of YAP, which modulates cell differentiation and proliferation [9,10]. The role of YAP in mechanotransduction relies on a specialized assembly of the F-actin cytoskeleton [8]. During respiration, alveolar cells are exposed to mechanical stress, which significantly affects homeostasis and the pathogenesis of pulmonary diseases [11]. The continuous inflation and deflation of the lungs induce the differentiation of type II pneumocytes into type I pneumocytes [12]. Mouse model studies revealed that reduced YAP expression in lungs during development causes severe impairment in airway patterning and lung regeneration after lung injury [13]. The gene amplification and epigenetic changes of YAP are widespread in many cancers [3]. In lung cancer, YAP functions as an oncogene by promoting cell proliferation and survival [3]. YAP also confers resistance to chemotherapeutic agents such as epidermal growth factor receptor (EGFR)-tyrosine kinase inhibitors (TKIs), and anaplastic lymphoma kinase (ALK) inhibitors in lung cancer [14,15,16]. Additionally, YAP activity influences the tumor microenvironment (TME) and directly regulates the transcription of programmed death-ligand 1 (PD-L1), which is an important target for immuno-oncological therapies [17,18,19]. Therefore, several research groups are investigating YAP modulation to defeat drug resistance and improve the effectiveness of immunotherapies in cancer patients [20]. 

Autophagy is a core cell survival mechanism involving the digestion of damaged organelles, malformed proteins, and unnecessary proteins as a reaction to cellular threat, nutrient deficiency, and chemotherapy [21,22]. Its functions differ depending on the physiological context. In normal cells, it inhibits carcinogenesis via the degradation of abnormal proteins [23]. However, it can accelerate the survival of cancer cells and induce resistance to chemotherapy in established tumors [23]. 

Organoids are novel three-dimensional (3D) multicellular structures derived from stem or progenitor cells, and they can potentially be used for applications in both basic and clinical research [24,25]. They can recapitulate the structures and some functions of specific organs. Currently, alveolar and airway organoids are actively used in various areas of lung research, including lung regeneration, lung cancer, and infectious lung diseases, including coronavirus disease-19 (COVID-19) [24,26,27]. Interestingly, several novel studies have suggested that YAP plays a vital role in the autophagy of lung cancer and the formation of lung organoids [28,29].

In this review, we deliberate the basic functions and recently discovered roles of YAP in chemo-resistance, tumor immunity, autophagy, and lung organoids, as well as the clinical implications of targeting YAP in lung cancer treatment.

## 2. Overview of Hippo Pathway and YAP in Lung Development and Regeneration

The Hippo pathway adjusts cell proliferation, differentiation, and organ extent during development and is involved in carcinogenesis, chemotherapeutic drug resistance, and tumor metastasis [30]. The canonical Hippo pathway comprises multiple components, including mammalian sterile 20-related kinases 1 and 2 (MST1/2), Salvador-1, large tumor suppressor kinase 1 and 2 (LATS1/2), mps1 binding proteins 1 and 2 (MOB1/2), transcriptional coactivator with PDZ-binding motif (TAZ), and YAP. YAP and its paralog TAZ are the final effectors of the Hippo signaling pathway and function as transcriptional complexes when bound to the transcriptional enhanced associate domain (TEAD). MST1/2 activates LATS1/2, which inhibits YAP/TAZ via phosphorylation and sequestration in the cytoplasm. YAP and TAZ share several functions but exhibit distinct roles in specific contexts. Unphosphorylated YAP can enter cell nuclei and control the transcription of numerous key factors related to cell propagation and anti-programmed cell death [5,31]. In addition to the canonical Hippo pathway, various other signals regulate the activity of YAP (Figure 1). For example, G-protein coupled receptors (GPCRs), RAS-RAF, and metabolic signals including byproducts from the mevalonate pathway, act as YAP regulators independently from the Hippo pathway [32,33].

The Hippo pathway performs critical roles in lung development and regeneration. Specifically, YAP adjusts proximal-distal modeling in airway progenitor cells by inducing Sox2 expression as a reaction to transforming growth factor-β (TGF-β) [34]. YAP-deficient mice exhibit hypoplastic lungs and severely disrupted branching morphogenesis [35]. YAP also contributes to lung fibrosis by upregulating extracellular-matrix-related genes and inducing the TGF-β/SMAD signaling pathway [36,37]. Furthermore, it is essential for alveolar epithelial regeneration after bacterial pneumonia; in its absence, lung regeneration is delayed due to the continuous activation of NF-κB-mediated inflammatory response and type II pneumocytes [13].

## 3. The Role of YAP in Chemo-Resistance and Metastasis

YAP functions as an oncogene in numerous solid cancers [38]. It is overexpressed in 60–70% of non-small cell lung cancers (NSCLCs), and amplification of the YAP copy number occurs in ~15% of squamous lung cell cancers [20,38]. In addition, its overexpression is positively correlated with poor clinical prognoses in lung cancer patients [20,38]. Recent studies have revealed that YAP promotes chemoresistance in conventional chemotherapies and targeted therapies [20]. YAP not only activates ERK and AXL signaling, but also induces epithelial–mesenchymal transition (EMT) against EGFR-TKI treatment [16]. Several studies have shown that YAP inhibition can overcome resistance to chemotherapy [39,40,41,42]. For example, the combined treatment of EGFR-TKI with YAP inhibitors suppresses EGFR-TKI resistance [16]. In addition, AXL was found to be a responsible factor in a third-generation EGFR-TKI, osimertinib-persistent cells [43]. Moreover, YAP1 promotes resistance to an ALK inhibitor, alectinib, by regulating pro-apoptotic proteins such as Mcl-1 and Bcl-xL. Co-treatment of YAP inhibitor suppresses the initial survival of cancer cells against alectinib [44]. In small-cell lung cancer (SCLC), YAP promotes multidrug resistance by triggering a cluster of differentiation (CD) 74 related signaling pathways [45]. It also induces epigenetic reprogramming in lung cancer, leading to tumor dormancy and a senescence-like state [46]. 

In the processes of EMT and metastasis, YAP is known to interact with many factors including AXL, β-catenin, and Slug [14,39,47]. In mice, YAP promotes metastasis in lung cancer by activating CD24+/Sca1+ tumor-propagating cells and inducing Slug transcription [48]. It also induces EMT by activating several downstream target genes, including forkhead box C2 (FOXC2), twist-related protein 1(TWIST), and zinc finger E-box binding homeobox 1 (ZEB1) [48,49]. Furthermore, YAP promotes metastasis by enhancing the extravasation of cancer cells and rendering circulating cancer cells resistant to anoikis [49]. It also regulates the transcription of Rho GTPase, which decreases cytoskeleton rigidity and enhances the metastatic phenotype of cancer cells [50]. When tumor cells metastasize into lymph nodes and adapt to the TME, they redirect their metabolism toward fatty acid oxidation; YAP activation accelerates these metabolic changes in cancer cells. Jin et al. discovered that norcantharidin, a demethylated form of cantharidin, reverses cisplatin resistance and inhibits EMT in NSCLC by regulating the YAP pathway [51]. In summary, these studies show that YAP inhibition might be a promising strategy to suppress metastasis of tumor cells. 

## 4. The Role of YAP in Tumor Immunity and Microenvironment

Tumor development and progression depend on the TME, which is composed of immune, endothelial, and other stromal cells [52]. YAP has immunomodulatory effects by regulating various types of immune cells [53]. For example, it decreases the differentiation of CD8β cells by hindering the transcription of B lymphocyte-induced maturation protein-1. Furthermore, its expression controls the immunosuppressive activity of T regulatory cells (Tregs) [54], highlighting its vital role in the function of Tregs and relation to the TGF-β/SMAD axis [54]. Macrophages control both innate and adaptive immunity and significantly affect the TME [55]. Depending on the conditions, macrophages can differentiate into classically activated macrophages (M1), which have anti-tumoral and proinflammatory functions, or activated macrophages (M2), which exhibit pro-tumoral and angiogenic tissue-remodeling functions. YAP expression can modulate macrophage differentiation and control the development and functionality of macrophages [56].

PD-L1 is an important protein target of immuno-oncological NSCLC treatments. Deepening our understanding of PD-L1 is essential to improving immunotherapies for NSCLC [57]. PD-L1 is regulated by various inherent and extrinsic factors [54,55]. For example, interferon-γ is the most prominent PD-L1-regulating extrinsic factor released by immune cells. Important intrinsic factors that adjust PD-L1 expression in cancer cells include the mechanistic target of rapamycin (mTOR), mitogen-activated protein kinase, and Myc [58,59]. Recent studies have found that YAP directly controls the transcription of PD-L1 [17,18] and that YAP inhibitors can modulate immune evasion by controlling PD-L1 expression [17,18]. Generally, PD-L1 is localized in the cell membrane and functions as a ligand of programmed death-1 (PD-1) [19]. However, a recent study revealed that nuclear PD-L1 has intrinsic functions that disrupt the effectiveness of anti-PD-1 immunotherapy. It demonstrated that PD-L1 nuclear translocation is controlled by acetylation and is associated with the efficacy of PD-1/PD-L1 targeted immunotherapy [60]. Furthermore, nuclear PD-L1 regulates several genes involved in immune responses, including NF-κB signal-related genes and major histocompatibility complex class I [59,60,61]. Smahel et al. showed that the presence of nuclear PD-L1 in circulating tumor cells is related to reduced survival rates in both prostate and colorectal cancer patients [62]. In another recent study, a screening for drug candidates capable of inhibiting PD-L1 expression identified verteporfin, a YAP inhibitor, as the most potent [63], suggesting that it can be repurposed as an adjuvant for immunotherapy.

In addition to the regulation of PD-L1, YAP alters the TME by recruiting and activating myeloid-derived suppressor cells via the upregulation of CXCL-5 in various solid cancers, such as prostate and pancreatic cancer [64]. YAP also plays a significant role in the differentiation of immune cells including Tregs, CD8+ T cells, T helper-17 cells, and macrophages [65,66]. In mouse tumor models, YAP deficiency in T cells reduces the recruitment of CD4+ and CD8+ T cells in tumors [65,66]. Based on these studies, YAP can be a novel target for controlling tumor immunity in lung cancer.

## 5. The Interplay between YAP and Autophagy

Autophagy is a physiologic process activated to survive against cellular stresses such as serum starvation or augmented metabolic demands [21,67]. Remarkably, in cancer, autophagy plays the opposite role depending on the context. Before the establishment of a tumor, autophagy functions as a tumor suppressor by degrading abnormal proteins. However, in the established cancer cells, autophagy confers survival mechanisms against chemotherapy and radiation therapy [68]. In many solid cancers, including lung cancer, autophagy activation is generally correlated with poor prognosis [69]. 

Several studies have shown that YAP has a close relationship with autophagy in cancer [69,70,71,72,73,74]. For instance, YAP causes cisplatin resistance by activating autophagy in ovarian cancer [70]. Similarly, it induces autophagic flux to improve cell survival in breast cancer cells upon nutrient deprivation [71]. In hepatocellular carcinoma, YAP constrains autophagy-dependent cellular death via the RAC1-reactive oxygen species-mTOR signaling [72]. Interestingly, mechanical stress can also activate autophagy flux and induce cell phenotype plasticity via YAP activation [73]. Furthermore, contact inhibition and YAP activation contribute to cell survival and proliferation by regulating autophagosome formation [74]. 

In lung cancer, autophagy plays a prominent role in cell proliferation, metastasis, and drug resistance. High expression of p62, a cargo protein of autophagosomes, correlates with tumor proliferation in NSCLC [75]. Autophagy also contributes to chemoresistance in NSCLC under hypoxic conditions [15,69]. EGFR-TKI-resistant NSCLC cells stimulate autophagy against EGFR-TKI, and treatment with the autophagic inhibitor chloroquine partially suppresses EGFR-TKI resistance [15]. In addition, a study showed that camptothecin induces autophagic activation in NSCLC, and co-treatment with autophagy inhibitor 3-methyladenine can increase apoptosis of cancer cells [76]. A recent study demonstrated that YAP regulates cell proliferation by activating autophagy and inhibiting the AKT/mTOR pathway in lung adenocarcinoma [77]. Another study revealed that YAP controls the expression of p62 in lung adenocarcinoma, and the YAP inhibitor verteporfin suppresses YAP, p62, and PD-L1 simultaneously [15]. Therefore, YAP and autophagy are promising targets for treating intractable drug resistance in lung cancer.

## 6. The Special Role of YAP in Lung Organoids

Organoids are specialized 3D multi-cellular, microtissues created from embryonic stem cells, induced pluripotent stem (iPS) cells, or organ-specific adult stem cells/progenitor cells [78,79]. Whereas spheroids contain only the same cell type, organoids are composed of multiple cell types [80]. Since organoids can effectively recapitulate the 3D structures and functions of organs, many scientists think that organoid abilities are among the most significant scientific advancements with many clinical applications [81]. Lung and intestinal organoids derived from cystic fibrosis patients are currently being used to develop cystic fibrosis transmembrane conductance regulator (CFTR) modulating therapies [82]. Organoids are typically cultured in laminin-rich extracellular matrices such as Matrigel or Cultrex basement membrane extracts derived from Engelbreth–Holm–Swarm tumors. Although these matrices mimic the complex extracellular TME, growth factors from them have not been accurately clarified. Although the absence of immune cells and vasculatures is a major drawback of organoid systems, we can culture lung organoids with some immune and mesenchymal cells in a transwell plate [83]. Furthermore, hemodynamic flow processes can be applied to organ-on-a-chip systems to replicate physiological conditions [84]. At present, organoids are widely used for disease modeling and drug screening in various biomedical fields [80,81].

Lung organoids are formed from lung stem cells or progenitor cells and can be classified into two types: airway organoids and alveolar organoids [85]. Airway organoids are generally derived from basal cells, whereas type II pneumocytes are applied to generate alveolar organoids with or without supporting stromal cells [86]. Human airway organoids consist of Trp63+/Krt5+ basal cells, Mucin 5 (Muc5) AC+/Muc5AB+ secretory goblet cells, and multi-ciliated cells [87]. Human alveolar epithelial (AEC) type 2 cells are defined as CD31-/CD45-/ EPCAM+/HTII280+ cells, and they can form alveolar organoids with the majority of AEC2s and few AEC1s [88]. Lung tissues created from embryonic stem cells and iPS cells can also be induced to differentiate into the airway or alveolar organoids [85]. Lung organoids are currently used to investigate lung development, fibrosis, restoration, and cancer [26,89]. In addition, they are a valuable platform for studying infectious diseases such as COVID-19 [26]. Biobanks of lung cancer organoid derived from patients’ tissue have been established and successfully employed for drug screening [90,91].

During the development of organoids into specialized 3D systems, each cell is subjected to mechanical stress [81,85]. The mechano-sensitive YAP protein plays decisive functions in cell proliferation and differentiation and functions as a key controller for organoid development [92,93]. For intestinal organoids, the development of complex multicellular asymmetric structures involves self-organization and breaking symmetricity; this process requires YAP signaling [94]. YAP inhibition by verteporfin significantly reduces the size and number of esophageal organoids [95]. Mechano-sensitive growth coordination by the integrin–Src family kinase–YAP pathway is also essential for liver organoid formation [96]. During the development of tubular structures in lung organoids, actomyosin contraction and YAP activation are important factors influencing the organoid size and cell proliferation [29]. Inhibition of retinoic acid production increases the size and enhances the differentiation of lung organoids through YAP activation, whereas verteporfin treatment decreases the size and number of lung organoids [97]. Further investigations into YAP and mechanistic factors in lung organoids are crucial to advancing the engineering of lung organoids.

## 7. Clinical Implication of YAP Targeting and Verteporfin in Lung Cancer

YAP plays a central role in lung cancer progression, metastasis, drug resistance, and immune evasion [17,20,64]. Therefore, it is a promising target for suppressing tumor progression and drug resistance [20]. Several YAP inhibitors have been discovered, including dasatinib, statin, pazopanib, verteporfin, and dobutamine [98,99]. Most drugs attenuate YAP-dependent transcription by inhibiting its nuclear translocation. More specifically, dasatinib suppresses YAP via the inhibition of Src family kinases, and statin inhibits YAP nuclear localization by blocking sterol regulatory element-binding proteins or the mevalonate pathway [100]. Furthermore, verteporfin has been actively studied as a YAP inhibitor for cancer treatment [100,101]. It is known to inhibit YAP activity via disruption of the YAP/TEAD complex formation [100,102]. Originally, verteporfin was permitted by the Food and Drug Administration as a photodynamic therapeutic agent. However, recent studies have revealed that verteporfin can inhibit cell proliferation and drug resistance in the absence of light [103,104]. In several solid cancers such as ovarian, pancreatic, and colon cancer, verteporfin effectively inhibits cancer cell proliferation, invasion, and chemoresistance [104]. 

Verteporfin can also reduce resistance to anti-cancer drugs. It restores sensitivity to EGFR-TKIs in lung adenocarcinoma with primary and acquired EGFR-TKI-resistance [16]. It also significantly reduces the activation of EMT-related signals and restores susceptibility to chemotherapy in NSCLCs with mesenchymal characteristics [16,105]. In SCLC, verteporfin increases the apoptosis rate of cancer cells that exhibit drug resistance to cisplatin and etoposide [106].

In terms of autophagy, PD-L1, and TME, verteporfin has several additional functions. For example, it can inhibit autophagy in cancer cells by blocking the expansion of phagophores and causing the oligomerization of the autophagic adaptor p62 [107]. It also inhibits PD-L1 expression through YAP inhibition and disruption of STAT1-IRF1-TRIM28 signaling [66]. In the TME, it activates cytotoxic T lymphocytes by inhibiting PD-L1 expression in tumors [66]. In hepatocellular carcinoma, verteporfin increases sensitivity to cisplatin by reducing the expression of YAP, PD-L1, and TGF-β [20]. Considering its anti-tumor effects, verteporfin shows great potential in treating intractable lung cancer by overcoming drug resistance and enhancing the efficacy of cancer immunotherapies. Although verteporfin is clinically used as a photosensitizer for photodynamic therapy, its off-target side effects and optimal dosage for anti-tumor effects should be considered. Recently developed drugs, including CA3, can potently and selectively interfere with the bond between YAP1 and TEAD [98,99]. These drugs may be valuable for targeting YAP in lung cancer. 

## 8. Concluding Remarks and Future Perspectives

Over the last decade, the characteristics of YAP in lung cancer have been extensively elucidated. Recent studies have revealed that YAP acts as an oncogene by regulating cell proliferation and apoptosis, but also controlling TME and autophagy. Notably, it directly regulates the transcription of PD-L1, which is a major target in immuno-oncology. Recent in vitro and mouse model experiments have unveiled novel functions of YAP, including drug resistance, metastasis, and immune escape in lung cancers. The lately discovered connection between YAP and autophagy has helped develop novel strategies to suppress drug resistance and metastasis in lung cancer. Nowadays, lung organoids have become an important platform for investigating lung cancer, fibrosis, and infectious diseases such as COVID-19. Lung cancer organoids can recapitulate the genomic and molecular characteristics of the cancers from which they are derived. Interestingly, YAP is essential for the formation of alveolar-like cellular arrangements and tubular shapes of airway cells in lung organoids. Finally, YAP inhibitors, including verteporfin, and CA3 have great potential as novel strategies for overcoming drug resistance and as adjuvants for immunotherapeutics (Figure 2). However, further clinical studies are vital to translate these promising findings to the clinical field.

## Figures and Tables

**Figure 1 cancers-13-03069-f001:**
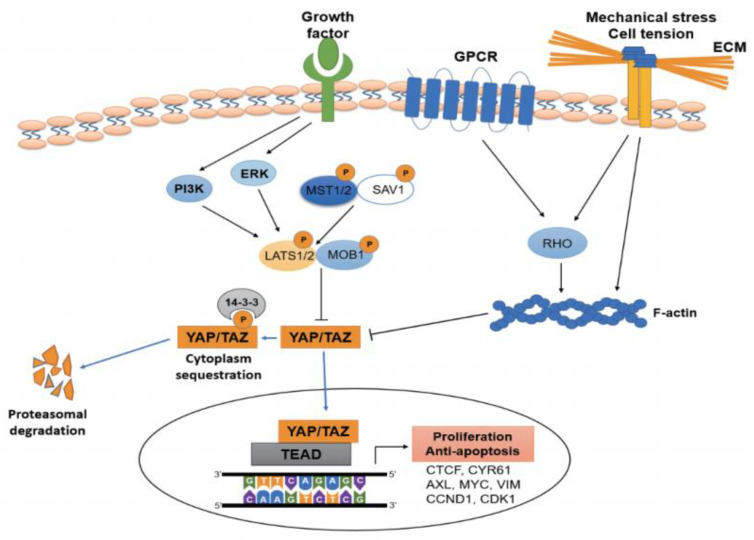
Core components and interacting signals of the Hippo/YAP signaling pathway in cancer. In the canonical Hippo pathway, Mst1/2 and Sav1 activate Lats1/2 and Mob1, which in turn phosphorylate YAP/TAZ. Then, phosphorylated YAP and TAZ are sequestered in the cytoplasm and degraded by proteasomes. Nuclear YAP/TAZ and TEAD complexes induce the transcription of multiple genes that regulate cell propagation and programmed cell death. Several growth factors, G-protein coupled receptors, and mechanical stimuli regulate the activation of YAP/TAZ.

**Figure 2 cancers-13-03069-f002:**
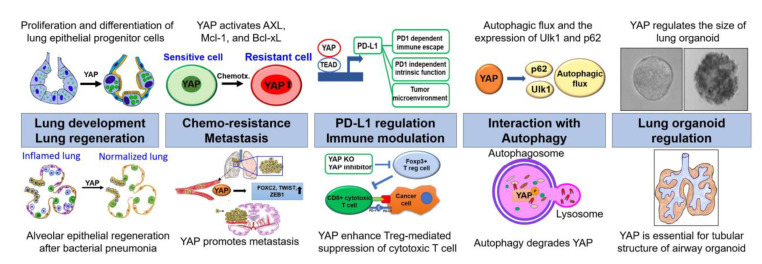
The crucial roles of YAP from chemoresistance and tumor immunity to autophagy and organoid. YAP is intimately involved in lung development and regeneration. In cancer, it promotes drug resistance by activating AXL, Mcl-1, and Bcl-xL. In addition, YAP induces the metastasis of cancer cells by increasing the expressions of FOXC2, TWIST, and ZEB1. It also directly regulates the transcription of PD-L1 and modulates the functions of several immune cell types. Furthermore, it regulates autophagic flux and the expression of several key autophagic factors, including p62 and Ulk1. Lastly, YAP plays a crucial role in regulating the size of lung organoids and developing the tubular structure of airway organoids.

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
