# Peer review of "New Insights into the Clinical Implications of Yes-Associated Protein in Lung Cancer: Roles in Drug Resistance, Tumor Immunity, Autophagy, and Organoid Development"

_cancers, 2021, doi:10.3390/cancers13123069_

Round 1

Reviewer 1 Report

In this review paper, the authors described the roles of YAP in lung cancer. The manuscript was well written and the important publications including the latest ones were cited. However, several amendments will be required for publication.

  • It is difficult to understand the role of YAP in “Chemo-resistance Metastasis” and “Lung organoid regulation” of Figure 2. The authors should modulate the figure 2 and state more explanations in the legend.
  • The section 5 “The interplay between YAP and Autophagy” is very hard to read and understand. The authors should rewrite the section and clearly indicate the dual function of autophagy in cancer.
  • Please write “Figure 1” and “Figure 2” in appropriate positions in the text.
  • The manuscript needs a thorough check on the words, abbreviations, references. In addition, some sentences do not sound right.

Some examples:

“type 2 pneumocyte” (line 106) → “type II pneumocyte”

“The role YAP” (line 109) → “The role of YAP”

“non-small cell lung cancers” (line 111) → “non-small cell lung cancers (NSCLC)”

“crizotinib” and “alectinib” (line 118 and line 120) → Please state what they are.

“induced pluripotent stem cells (iPS)” (line 210) → “induced pluripotent stem (iPS) cells”

“iPSs” (line 235) → “iPS cells”.

“localized YAP activation” (line 250) → “localized” is not required.

“it” (line 278 and line 279) → In the sentences, “it” is not clear what indicates.

The journal names are missing for reference number 9, 28, 36, 40, 41, 49, 53, 54, 60, 67, 74, 85, 86, 90, 100.

Author Response

Response to Reviewer 1 Comments

Point 1: It is difficult to understand the role of YAP in “Chemo-resistance Metastasis” and “Lung organoid regulation” of Figure 2. The authors should modulate the figure 2 and state more explanations in the legend.

Response 1: This is an excellent suggestion improving the paper’s quality. We’ve modulated the figure 2 and added some explanations in the figure legend.

: Figure 2, Figure legend

Point 2: The section 5 “The interplay between YAP and Autophagy” is very hard to read and understand. The authors should rewrite the section and clearly indicate the dual function of autophagy in cancer.

Response 2: We appreciated your helpful comments. We’ve written the section again and tried to clarify the dual function of autophagy in cancer. 

: Section 5. Line 191-197

Point 3: Please write “Figure 1” and “Figure 2” in appropriate positions in the text.

Response 3: Thank you very much. We’ve inserted “Figure 1.” and “Figure 2.” in the manuscript.

: section 2, Line 99, 104, 115, 137, 150, 170, 199, 220, and 259

Point 4: The manuscript needs a thorough check on the words, abbreviations, references. In addition, some sentences do not sound right.

Response 4: We appreciated the reviewer’s thoughtful comments. We’ve checked the manuscript thoroughly and revised it.

Point 5: Some examples:

“type 2 pneumocyte” (line 106) → “type II pneumocyte”: We’ve changed it.

“The role YAP” (line 109) → “The role of YAP”: We’ve changed it.

“non-small cell lung cancers” (line 111) → “non-small cell lung cancers (NSCLC)”

: We’ve changed it.

“crizotinib” and “alectinib” (line 118 and line 120) → Please state what they are.

 : We deleted the contents about crizotinib and stated alectinib as ALK inhibitor.

“induced pluripotent stem cells (iPS)” (line 210) → “induced pluripotent stem (iPS) cells”

: We’ve changed it.

“iPSs” (line 235) → “iPS cells”. : We’ve changed it.

“localized YAP activation” (line 250) → “localized” is not required. : We’ve deleted it.

“it” (line 278 and line 279) → In the sentences, “it” is not clear what indicates.

 : We replaced “it” to verteporfin to clarify the meaning of the sentence.

Point 5: The journal names are missing for reference number 9, 28, 36, 40, 41, 49, 53, 54, 60, 67, 74, 85, 86, 90, 100.

Response 5: We’ve corrected the references.

Reviewer 2 Report

I feel that this is an excellent and well-organized review, which describes many aspects of the relationship between YAP and lung cancer.
There are a few minor comments.

References 1-3 are from reviews, which I think is fine from a literary point of view . Why not cite a review that had a significant impact, such as (Nat Rev Cancer 19, 454-464 (2019), https://doi.org/10.1038/s41568-019-0168-y)?  If possible, it would be better to cite literature written in English, which can be widely read.

References 4.5 are from the review. Since this deals with individual events such as "Discovery of Yki in Drosophia", it is better to cite the original source. Please make sure that the review is appropriate for the other citations.

Line110-137: In recent years, in the field of molecular target drug resistance, not only acquired resistance but also phenomena such as "drug persistant cells" or "initial survival" have attracted attention. β-catenin and AXL was reported as responsible factor in drug persistant cells in EGFR-positve lung cancer  (Nat Commun 10, 259 (2019).  https://doi.org/10.1038/s41467-018-08074-0,). YAP1 and AXL have been highlighted as responsible factors in ALK positive lung cancer (Nat Commun 11, 74 (2020). https://doi.org/10.1038/s41467-019-13771-5, https://cancerres.aacrjournals.org/content/74/20/5878,). These factors (AXL, YAP, b-catenin) have also known as important factors that linked to EMT. I think it would be better if this point was also considered.

Line 117-119, Crizotinib is not widely used in clinical practice in the first line, and second and later generation ALK inhibitors such as Alectinib are used. There is a literature on YAP1 being associated with resistance to alectinib in patient-derived cell data (Nat Commun 11, 74 (2020).)

Line 257-286, Verteporfin is mentioned as a YAP inhibitor, but the concentration used in experimental mice is very high, and it is questionable whether it can be applied to humans. Inhibitors of YAP1 are considered to be difficult to make, but several new drugs have been reported, so it would be good to consider these as well.

1.  CA3:https://mct.aacrjournals.org/content/17/2/443

2. https://mct.aacrjournals.org/content/early/2021/04/13/1535-7163.MCT-20-0717

Author Response

Response to Reviewer 2 Comments

Point 1: 1. References 1-3 are from reviews, which I think is fine from a literary point of view. Why not cite a review that had a significant impact, such as (Nat Rev Cancer 19, 454-464 (2019), https://doi.org/10.1038/s41568-019-0168-y)?  If possible, it would be better to cite literature written in English, which can be widely read.  

Response 1: Thank you very much for your comment. We’ve modified the references as you recommended.

: Ref. 1, 2, 3

  1. Zanconato F, Cordenonsi M, Piccolo S. YAP and TAZ: a signalling hub of the tumour microenvironment. Nat Rev Cancer. 2019;19(8):454-64.
  2. Xie H, Wu L, Deng Z, Huo Y, Cheng Y. Emerging roles of YAP/TAZ in lung physiology and diseases. Life sciences. 2018;214:176-83.
  3. Moroishi T, Hansen CG, Guan KL. The emerging roles of YAP and TAZ in cancer. Nature reviews Cancer. 2015;15(2):73-9.

Point 2. References 4.5 are from the review. Since this deals with individual events such as "Discovery of Yki in Drosophia", it is better to cite the original source. Please make sure that the review is appropriate for the other citations.

Response 2:  We really appreciate your considerate comment. We’ve changed the references as you recommended.

: Ref. 4. 5

  1. Justice RW, Zilian O, Woods DF, Noll M, Bryant PJ. The Drosophila tumor suppressor gene warts encodes a homolog of human myotonic dystrophy kinase and is required for the control of cell shape and proliferation. Genes & development. 1995;9(5):534-46.
  2. Kango-Singh M, Nolo R, Tao C, Verstreken P, Hiesinger PR, Bellen HJ, et al. Shar-pei mediates cell proliferation arrest during imaginal disc growth in Drosophila. Development (Cambridge, England). 2002;129(24):5719-30.6.

Point 3. Line110-137: In recent years, in the field of molecular target drug resistance, not only acquired resistance but also phenomena such as "drug persistent cells" or "initial survival" have attracted attention. β-catenin and AXL was reported as responsible factor in drug persistant cells in EGFR-positive lung cancer  (Nat Commun 10, 259 (2019).  https://doi.org/10.1038/s41467-018-08074-0,). YAP1 and AXL have been highlighted as responsible factors in ALK positive lung cancer (Nat Commun 11, 74 (2020). https://doi.org/10.1038/s41467-019-13771-5, https://cancerres.aacrjournals.org/content/74/20/5878,). These factors (AXL, YAP, b-catenin) have also known as important factors that linked to EMT. I think it would be better if this point was also considered.

Response 3:  This is an excellent suggestion improving the paper’s quality. As you recommended, we’ve added these contents and references in the manuscript.

: section 3, line 122-128, 132-133   Ref) 14,44

Point 4. Line 117-119, Crizotinib is not widely used in clinical practice in the first line, and second and later generation ALK inhibitors such as Alectinib are used. There is a literature on YAP1 being associated with resistance to alectinib in patient-derived cell data (Nat Commun 11, 74 (2020).)

Response 4:  We appreciate the reviewer’s thoughtful comment. We’ve removed the content of crizotinib and added some detail related to alectinib in consideration of your suggestion. 

: section 3, line 125-126, Ref) 44

Point 5. Line 257-286, Verteporfin is mentioned as a YAP inhibitor, but the concentration used in experimental mice is very high, and it is questionable whether it can be applied to humans. Inhibitors of YAP1 are considered to be difficult to make, but several new drugs have been reported, so it would be good to consider these as well.

  1. CA3:https://mct.aacrjournals.org/content/17/2/443
  2. https://mct.aacrjournals.org/content/early/2021/04/13/1535-7163.MCT-20-0717

Response 5:  This is an excellent suggestion improving the paper’s quality. We’ve added the contents about novel specific YAP inhibitors in the manuscript and references as you recommended.

: Section 7, line 303-307 and Section 8, line 322 Ref) 98, 99

Reviewer 3 Report

The topic of the review is important and relevant to cancer. However, the authors just list different observations made in relation to YAP, but the mechanisms are not provided. It has been mentioned several times throughout the text that YAP regulates mechanotransduction. The mechanism, however, was not presented and discussed. It is not shown how the normal YAP function is utilised by cancer cells to promote tumour growth and progression.  Cooperation with other factors (such as for example AP1) is not discussed.

In addition:

Line 115, 116. “… induces epithelial-mesenchymal transition (EMT) against EGFR-TKI treatment”. Possibly style has to be corrected

Line 127 ”… snail family zinc finger 1 and 2”. Is this slug and snail? If yes, slug was just already discussed (line 125)

Line 128 “E-box-binding homeobox 1”.  Use more common names ZEB1 or for the gene ZFHX1A

Lines 140-151. In which cells, stromal or tumour, YAP is expressed?

What is the function of nuclear PD-L1? Is YAP important for the expression of membranous PD-L1 or nuclear or both?

201, 202. “YAP also regulates cell proliferation by activating autophagy and inhibiting the AKT/mTOR pathway in lung adenocarcinoma (73)”. This reference contains no information on AKT/mTOR or cell proliferation.

Line 204. Verteporfin is mentioned here for the first time but not explained.

Lines 283, 284. . “In hepatocellular carcinoma, it increases sensitivity to cisplatin by reducing the expression of YAP, PD-L1, and TGF-β (20)”. From this statement it looks as if YAP is reducing the expression of YAP. Not sure that this was meant by the authors.

Author Response

Response to Reviewer 3 Comments

The topic of the review is important and relevant to cancer. However, the authors just list different observations made in relation to YAP, but the mechanisms are not provided. It has been mentioned several times throughout the text that YAP regulates mechanotransduction. The mechanism, however, was not presented and discussed. It is not shown how the normal YAP function is utilised by cancer cells to promote tumour growth and progression.  Cooperation with other factors (such as for example AP1) is not discussed.

Point 1: The mechanism of YAP and mechanotransduction.

Response 1: We appreciate the reviewer’s thoughtful comment. The role YAP in mechanotransduction is known to rely on a specialized assembly of the F-actin cytoskeleton. We’ve added this content related to the mechanism in the introduction section.  

: Section 1, line 50, 51. Ref) 8

Point 2: how the normal YAP function is utilised by cancer cells to promote tumour growth and progression.

Response 2: Thank you very much for your comment. Several studies showed that the gene amplification and epigenetic changes of YAP is widespread in many cancers. YAP is known to function as an oncogene by activating the transcription of genes related to cell proliferation and survival. We’ve added this content in the manuscript.

: Section 1, line 57-59. Ref) 3

Point 3: Cooperation with other factors (such as for example AP1)

Response 3:  We appreciate the reviewer’s considerate comment. As you recommend, we’ve addressed the YAP’s cooperation partners such as AP1, β-catenin, and TGFβ.

: Section 1, line 45-47. Ref) 3, 7

Point 4: Line 115, 116. “… induces epithelial-mesenchymal transition (EMT) against EGFR-TKI treatment”. Possibly style has to be corrected

Response 4: Thank you for your comment. We’ve changed the style of the sentence.

: section 3, line 119

Point 5: Line 127 ”… snail family zinc finger 1 and 2”. Is this slug and snail? If yes, slug was just already discussed (line 125)

Response 5: We appreciate your considerate comment. We’ve deleted the “snail family zinc finger 1 and 2” in the manuscript.

Point 6: Line 128 “E-box-binding homeobox 1”.  Use more common names ZEB1 or for the gene ZFHX1A

Response 6: Thank you for your comment. We used the term “ZEB1”.

: section 3. Line 137

Point 7: Lines 140-151. In which cells, stromal or tumour, YAP is expressed?

Response 7: Thank you very much for your valuable comment. In this part, we had tried to emphasize the role of YAP in various immune cells. We’ve clarified the sentence.  “YAP has immunomodulatory effects by regulating various types of immune cells.”

: section 4, line 151

Point 8: What is the function of nuclear PD-L1? Is YAP important for the expression of membranous PD-L1 or nuclear or both?

Response 8: We appreciate your considerate comment. The recent paper showed that nuclear PD-L1 regulates several genes involved in immune responses, including NF-κB signal-related genes and major histocompatibility complex class I. We think that YAP might be important for the expression of PD-L 1 in both nuclear and membrane since YAP directly regulates the transcription of PD-L1.

Point 9: 201, 202. “YAP also regulates cell proliferation by activating autophagy and inhibiting the AKT/mTOR pathway in lung adenocarcinoma (73)”. This reference contains no information on AKT/mTOR or cell proliferation.

Response 9: Thank you very much for your valuable comment. We think there was a mistake in the order of references. We intended to quote “Hippo pathway critical transcriptional coactivators YAP manipulates the proliferation of lung adenocarcinoma, which is regulated by PTEN/AKT/mTOR autophagic signaling (77)”. We thoroughly reviewed the references again and corrected them. 

: section 5, line 212-214

Point 10:. Line 204. Verteporfin is mentioned here for the first time but not explained.

Response 10: We appreciate your considerate comment. We added the short explanation for verteporfin. The details of verteporfin are contained in section 7.

: section 5, line 218

Point 11:. Lines 283, 284. . “In hepatocellular carcinoma, it increases sensitivity to cisplatin by reducing the expression of YAP, PD-L1, and TGF-β (20)”. From this statement it looks as if YAP is reducing the expression of YAP. Not sure that this was meant by the authors.

Response 11: Thank you for your comment. We meant that in hepatocellular carcinoma, verteporfin increases sensitivity to cisplatin by reducing the expression of YAP, PD-L1, and TGF-β. We’ve clarified the sentence.

: section 7, line 299
